# REINVENT-Transformer: Molecular De Novo Design through Transformer-based Reinforcement Learning

Pengcheng Xu
px6@illinois.edu
Univeresity of Illinois Urbana Champaign
Urbana, Illinois, USA

Tianfan Fu
fut2@rpi.edu
Rensselaer Polytechnic Institute
Troy, New York, USA

Wenhao Gao
whgao@mit.edu
Massachusetts Institute of Technology
Boston, Massachusetts , USA

Jimeng Sun
jimeng@illinois.edu
Univeresity of Illinois Urbana Champaign
Urbana, Illinois, USA

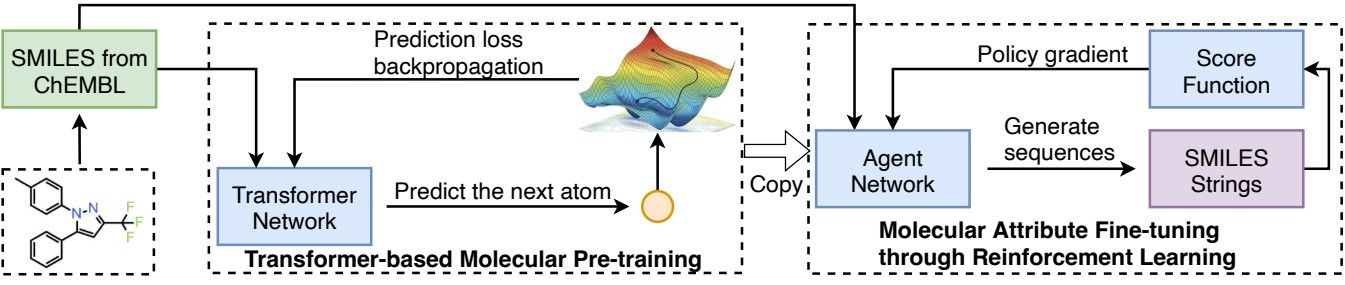

Figure 1: The framework of our method.

## ABSTRACT

In this work, we introduce a method: REINVENT-Transformer to fine-tune a Transformer-based generative model for molecular de novo design. Leveraging the superior sequence learning capacity of Transformers over Recurrent Neural Networks (RNNs), our model can generate molecular structures with desired properties effectively. In contrast to the traditional RNN-based models, our proposed method exhibits superior performance in generating compounds predicted to be active against various biological targets, capturing long-term dependencies in the molecular structure sequence. The model's efficacy is demonstrated across numerous tasks, including generating analogues to a query structure and producing compounds with particular attributes, outperforming the baseline RNN-based methods. Our approach can be used for scaffold hopping, library expansion starting from a single molecule and generating compounds with high predicted activity against biological targets.

## CCS CONCEPTS

• **Computing methodologies** → **Learning latent representations**; • **Applied computing** → *Biological networks.*

## KEYWORDS

Molecular Optimization, Transformer, Reinforcement Learning

**ACM Reference Format:**
Pengcheng Xu, Tianfan Fu, Wenhao Gao, and Jimeng Sun. 2024. REINVENT-Transformer: Molecular De Novo Design through Transformer-based Reinforcement Learning. In *Proceedings of Make sure to enter the correct conference title from your rights confirmation emai (Conference acronym 'XX).* ACM, New York, NY, USA, 16 pages. https://doi.org/XXXXXXX.XXXXXXX

## 1 INTRODUCTION

Navigating the vast chemical space, which contains $10^{60} - 10^{100}$ possible molecules, is a critical challenge in drug discovery [34, 45]. Early *de novo* design algorithms [6, 17] and RNN-based models [14, 32] have partially addressed this complexity. However, the Transformer architecture has proven superior, especially in handling the long-term dependencies necessary for modeling complex molecules, due to its:

(1) **Parallelization:** Processes all tokens simultaneously, unlike step-by-step processing in RNNs, enhancing efficiency.
(2) **Long-term Dependency Handling:** Employs multi-head self-attention mechanisms to capture long-range interactions.
(3) **Scalability:** Better suited for longer sequences, a key advantage in molecular design.

This work introduces a novel approach by incorporating the Decision Transformer in *de novo* molecular design. Leveraging "oracle feedback reinforcement learning," our model optimizes towards molecules with high predicted activity, providing precision and enhancing success rates in drug discovery [32]. This integration sets a new standard in the field, emphasizing the transformative potential of Transformer-based architectures in molecular design.

## 2 RELATED WORKS

Early *de novo* design algorithms primarily focused on structure-based methods, aiming to develop ligands that fit the binding pocket of a target, as highlighted in works by [6] and [17]. These methods often resulted in molecules with suboptimal drug metabolism and pharmacokinetic (DMPK) properties, presenting challenges in synthetic tractability. Ligand-based approaches, not relying on the 3D structure of the target, were introduced to address some of these limitations, involving a virtual library of chemical structures evaluated using a scoring function [21, 33]. However, the effectiveness of ligand-based methods compared to structure-based ones is not definitive [4].

Generative models, including RNN-based methods, have been employed in *de novo* design [10, 19, 35, 43]. These models learn the probability distribution over chemical structures and are further fine-tuned using reinforcement learning (RL) [24], achieving significant improvements.

The Transformer architecture, known for its self-attention mechanism, addresses the challenges of long-term dependencies in sequence data [39]. Motivated by its success, we propose using Transformer-based architectures for molecular *de novo* design in place of RNNs.

Molecular assembly strategies include string-based approaches like SMILES and SELFIES [29, 40], and graph-based methods representing molecular structures [26, 46]. Synthesis-based strategies focus on generating synthesizable molecules [7, 9, 10, 15].

Optimization algorithms used in molecular design include Genetic Algorithms (GAs) and Bayesian optimization (BO), which mimic natural evolutionary processes and build surrogates for the objective function, respectively [8, 28, 30, 31]. Variational autoencoders (VAEs) and reinforcement learning (RL) have also been used to map molecules from a latent space and refine models for enhanced molecule generation [19, 26, 32]. Recent advancements like Pasithea and Differentiable scaffolding tree (DST) utilize gradient-based optimization [11, 36].

The evolution from RNN-based methods to Transformer-based models reflects a desire to better handle complex chemical structures and optimize them more effectively. The Transformer architecture, particularly its self-attention mechanism, effectively handles sequence data [39]. Combined with RL for fine-tuning [12, 14, 24, 32, 42, 46], this integration aims to improve molecular design processes, promising more sophisticated automated systems.

## 3 METHODOLOGY: REINVENT-TRANSFORMER

Our method, named REINVENT-Transformer, first pre-trains the real 2D molecule dataset based on the transformer. Then, based on the RL paradigm, fine-tuning is performed on the molecular attributes to be optimized.

### 3.1 Preliminaries

This study focuses on single-objective molecular optimization for designing small organic molecules with significant scalar properties in therapeutic development. The molecular design task is an optimization problem:

$$m^* = \arg \max_{m \in \mathcal{M}} O(m),$$

where $m$ is a molecular structure and $\mathcal{M}$ is the vast chemical space of potential candidates, approximately $10^{60}$ [5]. We assume access to the actual value of a targeted property, $O(m) : \mathcal{M} \to \mathcal{R}$, evaluated by an oracle, $O$, which is an opaque mechanism providing a scalar value for specific chemical or biological attributes. These oracles, whether experimental or high-fidelity simulations, are costly. Hence, an efficient optimization algorithm within feasible resource constraints is vital, significantly aiding automated molecular design in advanced automated chemical design (ACD) [18] or function-driven autonomous synthesis [16].

### 3.2 Transformer-based Molecular Pre-training

The transformer is used for pre-training on real 2D molecules. Specifically, it treats the prediction of a 2D molecule as a sequence prediction and lets the transformer predict the next atom based on the molecular sequence history. The pre-training of the transformer is based on maximum likelihood.

**Training data Overview: Segmentation and Binary Coding of SMILES**

A Simplified Molecular Input Line Entry System (SMILES) [41] represents a molecule as a character sequence with atoms and symbols for ring closure, opening, and branching. SMILES are typically tokenized by single characters, except for two-character atoms like "Cl" and "Br" and special cases in square brackets (e.g., [nH]) treated as single tokens. This tokenization approach identified 86 tokens in the training data.

A molecule can have multiple SMILES representations. Canonicalization algorithms [40] ensure consistent SMILES for the same molecule, though different implementations may produce varied SMILES.

**Transformers Overview**  Transformers are a neural network architecture designed to process sequential data, while also accounting for the importance of each input in relation to the others, despite their position in the sequence [39]. They manage to do this by the introduction of an attention mechanism that assesses the significance of each input in the sequence (Figure 1). At any given step $t$, the transformer state at $t$ is influenced by all previous inputs $x_1, \ldots, x_{t-1}$ and the current input $x_t$. The transformer's ability to selectively focus on the parts of the input sequence that are most relevant for each step makes them especially well suited for tasks in the field of natural language processing. Sequences of words can be encoded into one-hot vectors with a length equivalent to our vocabulary size $X$. We may add two extra tokens, GO and EOS, to signify the beginning and end of a sequence, respectively.

**Learning to model the data** Training a Transformer for sequence modeling typically involves using maximum likelihood estimation to predict the next token $x_t$ in the target sequence, given tokens from the previous steps (Figure 1). The model generates a probability distribution at every step, representing the likely next character, and the objective is to maximize the likelihood assigned to the correct token:

$$J(\Theta) = -\sum_{t=1}^{T} \log P\left(x_t \mid x_{t-1}, \ldots, x_1\right). \quad (1)$$

The cost function $J(\Theta)$, often applied to a subset of all training examples known as a batch, is minimized with respect to the network parameters $\Theta$. Given a predicted log likelihood $\log P$ of the target at step $t$, the gradient of the cost function with respect to $\Theta$ is used to update $\Theta$. This method of fitting a neural network is called back-propagation. Changing the network parameters affects not only the immediate output at time $t$, but also influences the information flow into subsequent transformer states.

**Generating new samples** Once a Transformer has been trained on target sequences, it can be used to generate new sequences that adhere to the conditional probability distributions learned from the training set. The first input is the GO token, and at every timestep following, we sample an output token $x_t$ from the predicted probability distribution $P(X_t)$ over our vocabulary $X$. The sampled $x_t$ is then used as our next input. The sequence is considered finished once the EOS token is sampled.

## 3.3 Molecular Attribute Fine-tuning through Reinforcement Learning

In this part, we load the pre-trained transformer network and fine-tune it based on RL. Here, our task is to generate some specific molecules with good attributes. Therefore, we use the generated molecules to measure the properties of the corresponding molecules through Oracle, and use them as rewards to finetune the neural network.

**Agent Decision-Making and Markov Decision Processes** Consider an Agent choosing an action $a \in \mathbb{A}(s)$ in state $s \in \mathbb{S}$, where $\mathbb{S}$ is the set of states and $\mathbb{A}(s)$ is the set of actions. The policy $\pi(a \mid s)$ maps states to action probabilities. Reinforcement learning often uses Markov decision processes (MDPs), where the current state is sufficient for decision-making [37]. This can extend to partially observable MDPs with partial environment views. The reward $r(a \mid s)$ measures action effectiveness, and the long-term return $G(a_t, S_t) = \sum_t^T r_t$ is the cumulative reward from time $t$ to $T$. For molecular desirability, we consider the return of a complete SMILES sequence.

Reinforcement learning aims to improve the policy to maximize expected return $\mathbb{E}[G]$. Tasks ending at step $T$ are episodic [37], such as SMILES generation ending with an EOS token.

States and actions for training can be generated by the agent (on-policy) or others (off-policy) [37]. Two RL strategies are value-based and policy-based [37]. Value-based RL learns a value function to derive a policy, while policy-based RL directly learns the policy. For our problem, policy-based methods are preferred because:

- They can learn an optimal stochastic policy [37], aligning with our goal.
- Fine-tuning a prior sequence model with a scoring function needs minimal changes, reducing gradient estimate variance.

**Negative Log-Likelihood (NLL) and Loss Function** To assess the likelihood of sequence generation by the agent, we use the Negative Log-Likelihood (NLL). The NLL is calculated as follows:

$$NLL(S) = -\sum_{i=1}^{N} \ln P\left(X_i = T_i \mid X_{i-1} = T_{i-1} \ldots X_1 = x_1\right). \quad (2)$$

This measure is crucial in understanding the generative model's performance [3]. The augmented likelihood and loss function are then computed to adjust the agent's generation process:

$$NLL(S)_{\text{Augmented}} = NLL(S)_{\text{Prior}} - \sigma * MPO(S)_{\text{score}}$$

$$\text{loss} = \left[ NLL(S)_{\text{Augmented}} - NLL(S)_{\text{Agent}} \right]^2.$$

**Scoring Functions for Molecular Sequences** REINVENT-Transformer utilizes scoring functions to evaluate and guide the generation of molecular sequences. These functions are formulated as either a weighted product or a weighted sum:

$$S(x) = \left[ \prod_i p_i(x)^{w_i} \right]^{1/\sum_i w_i},$$

$$S(x) = \frac{\sum_i w_i * p_i(x)}{\sum_i w_i}.$$

This scoring approach is designed to balance various molecular properties during the generation process, facilitating the production of molecules with desired characteristics [3].

## 4 EXPERIMENT

### 4.1 Experimental Setup

*Dataset.* For methods requiring a database, we use the ZINC 250K dataset [22, 23], consisting of around 250K molecules. This dataset is significant in pharmaceuticals and is used in Screening [1], MolPAL [20], and pretraining generative models like VAEs [27] and LSTMs [44]. Essential fragments for JT-VAE [26], MIMOSA [13], and DST [11] are also derived from it.

*Baseline.* We compare eight baseline methods for performance evaluation, including REINVENT [32], Graph-GA [25], SELFIES-REINVENT [14], GP BO [38], STONED [31], SMILES-LSTM HC [8], SMILES-GA [8], SynNet [15], DoG-Gen [7], and DST [11]. The implementations come from the PMO benchmark [14][1].

*Metric.* In order to evaluate both optimization capability and sample efficiency, following [14], we use the area under the curve (AUC) of the top-$K$ average property value in relation to the number of oracle calls.

---

[1]https://github.com/wenhao-gao/mol_opt

## 4.2 Evaluation Results

Our result is shown in Table. 1. From the table, we can observe that our method is better than the baseline method on multiple Oracles, which proves the effectiveness of the transformer in our problem. Our experiments mainly follow the benchmark paper [14].

### Overall Molecular Generation Result

The evaluation results depict a thorough comparison between the REINVENT-Transformer (referred to as REINVENT-Trans) and other prominent models across multiple oracles. Randomly selected SMILES generated by different models can be seen in Table 4. And the corresponding chemical structures are shown in figure 5.

### Overall Molecular Generation Result Performance Overview

**REINVENT-Transformer** consistently excels in molecular generation, achieving top results in properties like 'Albuterol_Similarity', 'Mestranol_Similarity', 'QED', 'Scaffold_Hop', and 'Sitagliptin_MPO'. This indicates the model's strength in capturing intricate molecular patterns and optimizing desired properties.

### Comparative Insight

1. **Versus REINVENT (SMILES and SELFIES):** REINVENT-Transformer often outperforms REINVENT (SMILES), though REINVENT scores slightly better on 'Osimertinib_MPO'. SELFIES representation in REINVENT doesn't always enhance performance, highlighting the impact of model architecture and representations.

2. **Graph-based Models:** 'Graph GA' and 'GP BO' show strong performance in 'Amlodipine_MPO' and 'Celecoxib_Rediscovery' oracles, respectively, but aren't consistently top-performing, suggesting their limited generalizability.

3. **Genetic Algorithms:** STONED (using SELFIES) achieves the highest score in 'Fexofenadine_MPO', demonstrating the potential of genetic algorithms in specific optimization tasks despite their stochastic nature.

## 4.3 Ablation Study: Long Sequence Molecule Generation Comparison with REINVENT-SMILES

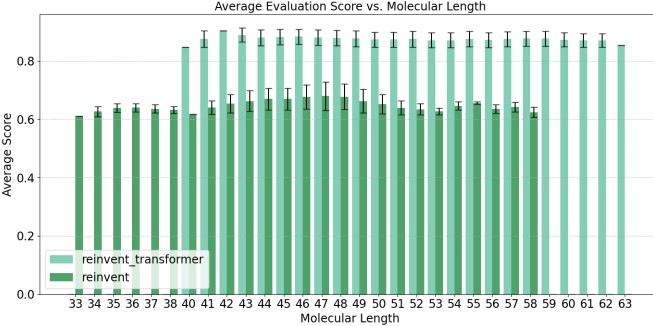

**Figure 2: Evaluation score vs molecular length for comparison of REINVENT-Transformer and REINVENT on oracle Mestranol_Similarity**

In order to better investigate in the ability of our method in long sequence generation, we did the following ablation study.

The box plot visualizes the distribution of evaluation scores across different molecular lengths for both the REINVENT-Transformer method and the baseline REINVENT method.

Based on the figure 2, we can derive the following observations:

1. In general, REINVENT-Transformer will generate longer average length of molecules than REINVENT.

2. The REINVENT-Transformer method consistently achieves higher average scores.

3. The spread (interquartile range) of scores for the REINVENT-Transformer method remains relatively consistent across molecular lengths, indicating stable performance.

In conclusion, the REINVENT-Transformer method outperforms the baseline REINVENT method, particularly in the context of longer molecular sequences.

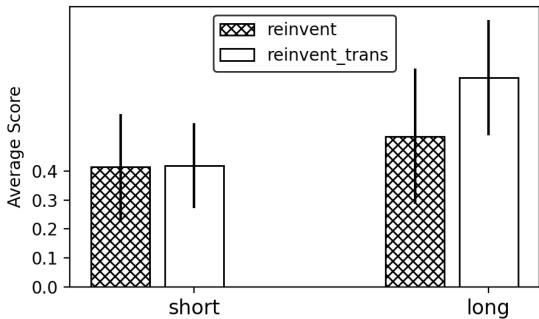

**Figure 3: Evaluation score vs short and long sequence for comparison of REINVENT-Transformer and REINVENT on oracle Mestranol_Similarity**

We set a threshold=50 for the length of the generated molecular string. If the generated string is longer than the threshold, it will be considered as "long", other it's considered as "short". From Figure 3, we can see the our method REINVENT-Transformer has better average score when generating long sequences.

## 4.4 Case Study: Convergence rate Comparison between REINVENT-Transformer and REINVENT

We plotted the auc_topk curve and the number of oracle calls is the x-axis. From the figure as follows, we can see that our method REINVENT-Transformer converges faster than REINVENT method.

From Fig. 4, the evolution of the average accuracy for the top 100 predictions is evident. Upon examination, across equivalent number of oracle calls, the mean accuracy of REINVENT-Transformer consistently surpasses that of REINVENT. This indicates a more expedient convergence rate for the REINVENT-Transformer compared to REINVENT.

The avg_top100 curve initially displays a steep incline, eventually plateauing post approximately 6000 oracle calls. Notably, beginning from the 2500th oracle call, the performance differential between REINVENT-Transformer and REINVENT significantly widens.

**Table 1: Performance comparison between REINVENT-Transformer, REINVENT, and other methods over all oracles for AUC Top-10**

| Method Assembly | REINVENT-Trans SMILES | REINVENT SMILES | Graph GA Fragments | REINVENT SELFIES | GP BO Fragments | STONED SELFIES |
|---|---|---|---|---|---|---|
| Albuterol_Similarity | **0.910± 0.008** | 0.882± 0.006 | 0.838± 0.016 | 0.826± 0.030 | 0.898± 0.014 | 0.745± 0.076 |
| Amlodipine_MPO | 0.653± 0.029 | 0.635± 0.035 | **0.661± 0.020** | 0.607± 0.014 | 0.583± 0.044 | 0.608± 0.046 |
| Celecoxib_Rediscovery | 0.457± 0.071 | 0.713± 0.067 | 0.630± 0.097 | 0.573± 0.043 | **0.723± 0.053** | 0.382± 0.041 |
| DRD2 | 0.931± 0.006 | 0.945± 0.007 | 0.964± 0.012 | 0.943± 0.005 | 0.923± 0.017 | 0.913± 0.020 |
| Deco_Hop | 0.645± 0.038 | 0.666± 0.044 | 0.619± 0.004 | 0.631± 0.012 | 0.629± 0.018 | 0.611± 0.008 |
| Fexofenadine_MPO | 0.796± 0.007 | 0.784± 0.006 | 0.760± 0.011 | 0.741± 0.002 | 0.722± 0.005 | **0.797± 0.016** |
| Isomers_C9H10N2O2PF2Cl | 0.809± 0.040 | 0.642± 0.054 | 0.719± 0.047 | 0.733± 0.029 | 0.469± 0.180 | 0.805± 0.031 |
| Median 1 | 0.354± 0.008 | **0.356± 0.009** | 0.294± 0.021 | 0.355± 0.011 | 0.301± 0.014 | 0.266± 0.016 |
| Median 2 | 0.263± 0.006 | 0.276± 0.008 | 0.273± 0.009 | 0.255± 0.005 | **0.297± 0.009** | 0.245± 0.032 |
| Mestranol_Similarity | **0.685± 0.032** | 0.618± 0.048 | 0.579± 0.022 | 0.620± 0.029 | 0.627± 0.089 | 0.609± 0.101 |
| Osimertinib_MPO | 0.813± 0.010 | **0.837± 0.009** | 0.831± 0.005 | 0.820± 0.003 | 0.787± 0.006 | 0.822± 0.012 |
| Perindopril_MPO | 0.525± 0.011 | 0.537± 0.016 | 0.538± 0.009 | 0.517± 0.021 | 0.493± 0.011 | 0.488± 0.011 |
| QED | **0.942± 0.000** | 0.941± 0.000 | 0.940± 0.000 | 0.940± 0.000 | 0.937± 0.000 | 0.941± 0.000 |
| Ranolazine_MPO | 0.761± 0.012 | 0.742± 0.009 | 0.728± 0.012 | 0.748± 0.018 | 0.735± 0.013 | **0.765± 0.029** |
| Scaffold_Hop | **0.560± 0.013** | 0.536± 0.019 | 0.517± 0.007 | 0.525± 0.013 | 0.548± 0.019 | 0.521± 0.034 |
| Sitagliptin_MPO | **0.563± 0.025** | 0.451± 0.003 | 0.433± 0.075 | 0.194± 0.121 | 0.186± 0.055 | 0.393± 0.083 |
| Thiothixene_Rediscovery | 0.556± 0.016 | 0.534± 0.013 | 0.479± 0.025 | 0.495± 0.040 | **0.559± 0.027** | 0.367± 0.027 |
| Troglitazone_Rediscovery | **0.451± 0.015** | 0.441± 0.032 | 0.390± 0.016 | 0.348± 0.012 | 0.410± 0.015 | 0.320± 0.018 |
| Valsartan_Smarts | **0.165± 0.278** | 0.165± 0.358 | 0.000± 0.000 | 0.000± 0.000 | 0.000± 0.000 | 0.000± 0.000 |
| Zaleplon_MPO | **0.544 ± 0.041** | 0.358± 0.062 | 0.346± 0.032 | 0.333± 0.026 | 0.221± 0.072 | 0.325± 0.027 |
| sum | 12.197 | 12.047 | 11.526 | 11.092 | 11.152 | 10.598 |
| rank | 1 | 2 | 3 | 5 | 4 | 6 |

| Method Assembly | LSTM HC SMILES | SMILES GA SMILES | SynNet Synthesis | DoG-Gen Synthesis | DST Fragments | |
|---|---|---|---|---|---|---|
| Albuterol_similarity | 0.719± 0.018 | 0.661± 0.066 | 0.584± 0.039 | 0.676± 0.013 | 0.619± 0.020 | |
| Amlodipine_MPO | 0.593± 0.016 | 0.549± 0.009 | 0.565± 0.007 | 0.536± 0.003 | 0.516± 0.007 | |
| Celecoxib_Rediscovery | 0.539± 0.018 | 0.344± 0.027 | 0.441± 0.027 | 0.464± 0.009 | 0.380± 0.006 | |
| DRD2 | 0.919± 0.015 | 0.908± 0.019 | **0.969± 0.004** | 0.948± 0.001 | 0.820± 0.014 | |
| Deco_Hop | **0.826± 0.017** | 0.611± 0.006 | 0.613± 0.009 | 0.800± 0.007 | 0.608± 0.008 | |
| Fexofenadine_MPO | 0.725± 0.003 | 0.721± 0.015 | 0.761± 0.015 | 0.695± 0.003 | 0.725± 0.005 | |
| Isomers_C9H10N2O2PF2Cl | 0.342± 0.027 | **0.860± 0.065** | 0.241± 0.064 | 0.199± 0.016 | 0.458± 0.063 | |
| Median 1 | 0.255± 0.010 | 0.192± 0.012 | 0.218± 0.008 | 0.217± 0.001 | 0.232± 0.009 | |
| Median 2 | 0.248± 0.008 | 0.198± 0.005 | 0.235± 0.006 | 0.212± 0.000 | 0.185± 0.020 | |
| Mestranol_Similarity | 0.526± 0.032 | 0.469± 0.029 | 0.399± 0.021 | 0.437± 0.007 | 0.450± 0.027 | |
| Osimertinib_MPO | 0.796± 0.002 | 0.817± 0.011 | 0.796± 0.003 | 0.774± 0.002 | 0.785± 0.004 | |
| Perindopril_MPO | 0.489± 0.007 | 0.447± 0.013 | **0.557± 0.011** | 0.474± 0.002 | 0.462± 0.008 | |
| QED | 0.939± 0.000 | 0.940± 0.000 | 0.941± 0.000 | 0.934± 0.000 | 0.938± 0.000 | |
| Ranolazine_MPO | 0.714± 0.008 | 0.699± 0.026 | 0.741± 0.010 | 0.711± 0.006 | 0.632± 0.054 | |
| Scaffold_Hop | 0.533± 0.012 | 0.494± 0.011 | 0.502± 0.012 | 0.515± 0.005 | 0.497± 0.004 | |
| Sitagliptin_MPO | 0.066± 0.019 | 0.363± 0.057 | 0.025± 0.014 | 0.048± 0.008 | 0.075± 0.032 | |
| Thiothixene_Rediscovery | 0.438± 0.008 | 0.315± 0.017 | 0.401± 0.019 | 0.375± 0.004 | 0.366± 0.006 | |
| Troglitazone_Rediscovery | 0.354± 0.016 | 0.263± 0.024 | 0.283± 0.008 | 0.416± 0.019 | 0.279± 0.019 | |
| Valsartan_Smarts | 0.000± 0.000 | 0.000± 0.000 | 0.000± 0.000 | 0.000± 0.000 | 0.000± 0.000 | |
| Zaleplon_MPO | 0.206± 0.006 | 0.334± 0.041 | 0.341± 0.011 | 0.123± 0.016 | 0.176± 0.045 | |
| sum | 10.227 | 10.185 | 9.613 | 9.554 | 9.203 | |
| rank | 7 | 8 | 9 | 10 | 11 | |

It is also observed that the REINVENT-Transformer possesses a higher standard deviation relative to REINVENT, suggesting potential variability in its performance. Despite this, the difference between the average top100 accuracy and the standard deviation for REINVENT-Transformer remains superior to the mean accuracy of REINVENT, reaffirming the enhanced efficacy of the REINVENT-Transformer method.

## 5 CONCLUSION

Navigating the vast chemical space in molecular design remains challenging, but the introduction of the REINVENT-Transformer marks a significant advancement by harnessing strengths such as parallelization and long-term dependency handling in the Transformer architecture. Our experimental findings demonstrate its superior performance across multiple oracles, particularly in tasks

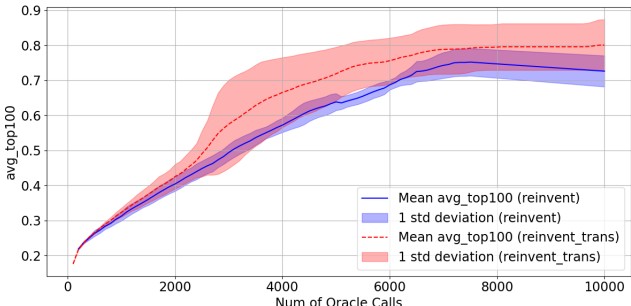

**Figure 4: Mean and Standard Deviation of avg_top100 over oracle calls for REINVENT and REINVENT-Transformer on oracle Mestranol_Similarity**

requiring longer sequence data, and integrating oracle feedback reinforcement learning enhances precision, favorably impacting drug discovery efforts. Ultimately, the REINVENT-Transformer sets a new benchmark in molecular de novo design and highlights the transformative potential of Transformer-based architectures in drug discovery.

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

# A  DETAILED INTRODUCTION

The vast expanse of chemical space, encompassing an order of magnitude from $10^{60} - 10^{100}$ possible synthetically feasible molecules [34, 45], presents formidable obstacles to drug discovery endeavors. In this colossal landscape, the task of pinpointing a molecule that simultaneously meets the prerequisites for bioactivity, drug metabolism and pharmacokinetic (DMPK) profile, and synthetic accessibility becomes an undertaking similar to the proverbial search for a needle in a haystack. Pioneering *de novo* design algorithms [6, 17] have attempted to address this by employing virtual strategies to design and evaluate molecules, thereby condensing the vast chemical space into a more navigable realm for exploration.

Traditional *de novo* molecule design models, based on Recurrent Neural Networks (RNNs), have proven effective in molecule generation tasks. However, RNNs possess inherent architectural limitations, notably in their capability to capture long-term dependencies in sequential data, which can be particularly detrimental when modeling complex molecular structures. Recently, the Transformer architecture has emerged as a powerful alternative to RNNs in sequence modeling tasks across various domains. Some of the key advantages of Transformers over RNNs include:

(1) **Parallelization:** Unlike RNNs which process sequences step-by-step, Transformers process all tokens in the sequence simultaneously, allowing for better computational efficiency.
(2) **Long-term Dependency Handling:** Transformers utilize multi-head self-attention mechanisms, which can capture long-range interactions in the data, making them particularly well-suited for modeling intricate molecular structures.
(3) **Scalability:** Transformers are inherently more scalable, allowing for the processing of longer sequences, which is a considerable advantage in molecular design.

In light of these advantages, our work introduces a novel approach by integrating the Transformer architecture, specifically the Decision Transformer, for *de novo* molecular design. By leveraging the inherent strengths of Transformers, our model exhibits enhanced performance in generating molecular structures with desired attributes.

Furthermore, we emphasize the incorporation of the "oracle feedback reinforcement learning" method. Pretraining models on large datasets is beneficial, but downstream tasks often require fine-tuning on specific objectives. By integrating feedback from an oracle during the reinforcement learning phase, our approach can efficiently navigate the solution space, optimizing towards molecules with high predicted activity. Such oracle-guided optimization provides an added layer of precision, facilitating the generation of molecules that not only conform to structural constraints but also exhibit high bioactivity, thereby increasing the potential success rate in drug discovery endeavors.

Drawing inspiration from previous work that employed RNNs and reinforcement learning for molecular optimization [32], our approach distinguishes itself by the adoption and fine-tuning of the Transformer architecture, ensuring superior handling of long-sequence data and paving the way for innovative breakthroughs in the realm of molecular design.

In summary, this work presents a fresh perspective on molecular *de novo* design, underscoring the potential of Transformer-based architectures, complemented by oracle feedback reinforcement learning, to revolutionize drug discovery methodologies. We envision that our approach will not only set a new benchmark in molecular generation tasks but will also inspire future research in leveraging advanced machine learning architectures for complex scientific challenges.

# B  DETAILED RELATED WORKS

Early *de novo* design algorithms primarily focused on structure-based methods, aiming to develop ligands that precisely fit the binding pocket of a target [6, 17]. While effective in certain aspects, these methods often resulted in molecules with suboptimal drug metabolism and pharmacokinetic (DMPK) properties and posed challenges in synthetic tractability. Ligand-based approaches were introduced to overcome some of these limitations, involving the creation of comprehensive virtual libraries of chemical structures evaluated using scoring functions [21, 33]. However, as noted by [4], the effectiveness of ligand-based methods compared to structure-based ones is not definitive, with both approaches having unique advantages and limitations depending on the specific requirements of the drug design process.

Recently, generative models such as RNN-based methods have been successfully applied to *de novo* design of molecules [10, 19, 35, 43]. These models have shown promise in learning the underlying probability distribution over large sets of chemical structures, effectively reducing the search space to reasonable molecules. Further improvements were achieved through fine-tuning using reinforcement learning (RL) techniques [24], demonstrating considerable enhancements over initial models.

Despite these advancements, challenges such as capturing long-term dependencies in sequence data persist. The Transformer architecture [39], known for its self-attention mechanism and ability to handle long sequences, has been highly successful in various sequence prediction tasks across domains. Motivated by these successes, we propose the use of Transformer-based architectures in place of RNNs for molecular *de novo* design.

Molecular assembly strategies play a crucial role in representing and manipulating chemical structures. String-based approaches like SMILES and SELFIES [29, 40] provide efficient representations of molecules. Graph-based methods offer intuitive two-dimensional representations of molecular structures, with nodes and edges representing atoms and bonds, respectively [26, 46]. Synthesis-based strategies aim to generate only synthesizable molecules, ensuring that the design aligns with real-world applications [7, 9, 10, 15].

Various optimization algorithms have been utilized for molecular design. Genetic Algorithms (GAs) mimic natural evolutionary processes and have been applied in molecule generation using both SMILES and SELFIES representations [8, 31]. Bayesian optimization (BO) builds a surrogate for the objective function, with applications such as BOSS and ChemBO in the molecular domain [28, 30]. Variational autoencoders

(VAEs) offer a generative approach, mapping molecules to and from a latent space, with notable methods including SMILES-VAE and JT-VAE [19, 26]. Reinforcement Learning (RL) techniques, like REINVENT, have also been applied to tune models for molecule generation [32]. Recent advancements in gradient ascent methods, such as Pasithea and Differentiable scaffolding tree (DST), have leveraged gradient-based optimization for molecular design [11, 36].

The evolution of molecular design methodologies has progressively addressed various challenges and limitations. The transition from RNN-based methods to more advanced generative models underscores a quest for improved handling of complex chemical structure representations and optimization. While RNNs brought significant progress, their inherent difficulty in capturing long-term dependencies in sequential data has been a notable shortcoming. The Transformer architecture addresses this gap through its self-attention mechanism, allowing for more nuanced and effective handling of sequence data, which is critical in molecular design where long-range interactions within molecules play a pivotal role.

The integration of reinforcement learning (RL) for fine-tuning generative models has further enhanced the field [24]. RL's ability to iteratively improve models based on a feedback loop aligns well with the demands of molecular design, where continuous refinement based on molecular properties is essential. The combination of RL with generative models has been shown to enhance the ability to navigate the vast chemical space more effectively, achieving better results in molecule generation [12, 14, 32, 42, 46].

In light of these advancements and existing limitations, our work proposes an approach that integrates the Transformer architecture with advanced RL techniques. This proposal is underpinned by the Transformer's superior handling of sequential data and the iterative refinement capability of RL. By merging these two powerful technologies, we aim to address the existing challenges in molecular *de novo* design, such as the need for better sequence representation and optimization. This integration promises to enhance the effectiveness and efficiency of molecular generation processes, moving closer to achieving more sophisticated and automated molecular design systems.

## C TRANSFORMER OVERVIEW FORMULA

A transformer block is a parameterized function class $f_\theta : \mathbb{R}^{n \times d} \to \mathbb{R}^{n \times d}$. If $\mathbf{x} \in \mathbb{R}^{n \times d}$ then $f_\theta(\mathbf{x}) = \mathbf{z}$ where

$$Q^{(h)}(\mathbf{x}_t) = W_{h,q}^T \mathbf{x}_t, \quad K^{(h)}(\mathbf{x}_t) = W_{h,k}^T \mathbf{x}_t,$$
$$V^{(h)}(\mathbf{x}_t) = W_{h,v}^T \mathbf{x}_t, \quad W_{h,q}, W_{h,k}, W_{h,v} \in \mathbb{R}^{d \times k} \tag{3}$$

$$\alpha_{t,j}^{(h)} = \text{softmax}_j \left( \frac{\left\langle Q^{(h)}(\mathbf{x}_t), K^{(h)}(\mathbf{x}_j) \right\rangle}{\sqrt{k}} \right), \tag{4}$$
$$\text{for } j = 1, \dots, t$$

$$\mathbf{u}_t' = \sum_{h=1}^{H} W_{c,h}^T \sum_{j=1}^{t} \alpha_{t,j}^{(h)} V^{(h)}(\mathbf{x}_j), \qquad W_{c,h} \in \mathbb{R}^{k \times d} \tag{5}$$

$$\mathbf{u}_t = \text{LayerNorm}\left(\mathbf{x}_t + \mathbf{u}_t'; \gamma_1, \beta_1\right), \qquad \gamma_1, \beta_1 \in \mathbb{R}^d \tag{6}$$

$$\mathbf{z}_t' = W_2^T \text{ReLU}\left(W_1^T \mathbf{u}_t\right), \quad W_1 \in \mathbb{R}^{d \times m}, W_2 \in \mathbb{R}^{m \times d} \tag{7}$$

$$\mathbf{z}_t = \text{LayerNorm}\left(\mathbf{u}_t + \mathbf{z}_t'; \gamma_2, \beta_2\right), \qquad \gamma_2, \beta_2 \in \mathbb{R}^d \tag{8}$$

$$\hat{\mathbf{y}} = \text{softmax}\left(W_z^T \mathbf{z}\right) = \frac{\exp\left(W_z^T \mathbf{z}\right)}{\sum_{k=1}^{m} \exp\left(W_z^T \mathbf{z}\right)_k}, \quad W_z \in \mathbb{R}^{d \times o}. \tag{9}$$

The notation $\text{softmax}_j$ indicates we take the softmax (defined in Equation 9) over the $d$-dimensional vector indexed by $j$. The LayerNorm function [2] is defined for $\mathbf{z} \in \mathbb{R}^k$ by

$$\text{LayerNorm}(\mathbf{z}; \gamma, \beta) = \gamma \frac{(\mathbf{z} - \mu_{\mathbf{z}})}{\sigma_{\mathbf{z}}} + \beta, \quad \gamma, \beta \in \mathbb{R}^k \tag{10}$$

$$\mu_{\mathbf{z}} = \frac{1}{k} \sum_{i=1}^{k} \mathbf{z}_i, \quad \sigma_{\mathbf{z}} = \sqrt{\frac{1}{k} \sum_{i=1}^{k} (\mathbf{z}_i - \mu_{\mathbf{z}})^2}. \tag{11}$$

The set of parameters, denoted by $\theta$, comprises the elements of the weight matrices $W$ and the LayerNorm parameters $\gamma$ and $\beta$, as specified on the right-hand side. The input $\mathbf{x} \in \mathbb{R}^{n \times d}$ represents a set of $n$ entities, each characterized by $d$ attributes (typically, though not exclusively, sequences of $d$-dimensional vectors of length $n$). It is important to note that the output $\mathbf{z} \in \mathbb{R}^{n \times d}$ retains the same format as the input $\mathbf{x} \in \mathbb{R}^{n \times d}$. A transformer is an amalgamation of $L$ distinct transformer blocks, each equipped with unique parameters: $f_{\theta_L} \circ \cdots \circ f_{\theta_1}(\mathbf{x}) \in \mathbb{R}^{n \times d}$.

Key hyperparameters in a transformer include $d, k, m, H$, and $L$, with typical configurations being $d = 512, k = 64, m = 2048, H = 8$. While the initial research suggested $L = 6$, more recent studies tend to employ a greater number of these blocks.

## D  DETAILED METHOD

### Agent Decision-Making and Markov Decision Processes

We frame the problem of generating a SMILES representation of a molecule with specified desirable properties via a Transformer as a partially observable Markov decision process. In this framework, an Agent must decide on an action $a \in \mathbb{A}(s)$ to take given a particular state $s \in \mathbb{S}$, where $\mathbb{S}$ denotes the set of possible states and $\mathbb{A}(s)$ represents the set of potential actions for that state. The policy $\pi(a \mid s)$ of an Agent associates a state to the likelihood of each action executed within.

Many reinforcement learning problems are modeled as Markov decision processes, indicating that the current state provides all essential information to inform our action choice, with no additional benefit from knowing past states' history. While this is more of an approximation than a fact for most real-life challenges, we extend this concept to a partially observable Markov decision process where the Agent interacts with a partial environment representation.

Let $r(a \mid s)$ be the reward serving as an indicator of the effectiveness of an action taken at a certain state, and the long-term return $G(a_t, S_t) = \sum_t^T r_t$ represents the cumulative rewards collected from time $t$ to time $T$ [37]. As molecular desirability is only meaningful for a completed SMILES, we will only consider a complete sequence's return.

The main objective of reinforcement learning is to enhance the Agent's policy to increase the expected return $\mathbb{E}[G]$ based on a set of actions taken from some states and the obtained rewards. A task with a definitive endpoint at step $T$ is known as an episodic task [37], where $T$ corresponds to the episode's length. SMILES generation is an example of an episodic task, which concludes once the EOS token is sampled.

The states and actions used for Agent training can be produced by the agent itself or through other means. If the agent generates them, the learning is called on-policy, and if generated by other means, it is off-policy learning [37].

Reinforcement learning commonly employs two different strategies to determine a policy: value-based RL and policy-based RL [37]. In value-based RL, the aim is to learn a value function that describes a given state's expected return. Once this function is learned, a policy can be established to maximize a certain action's expected state value. In contrast, policy-based RL aims to learn a policy directly.

For the problem we are addressing, we believe policy-based methods are the most suitable for the following reasons:

- Policy-based methods can explicitly learn an optimal stochastic policy [37], which aligns with our objective.
- Our method starts with a prior sequence model. The goal is to fine-tune this model based on a specific scoring function. Since the prior model already embodies a policy, fine-tuning might require only minimal changes to the prior model.
- The episodes in our case are short and fast to sample, reducing the impact of the variance in the estimate of the gradients.

In our approach, we use the probability distributions learned by a pre-trained Transformer model as our initial prior policy. We refer to the network using the prior policy simply as the Prior, and the network whose policy has been modified as the Agent. The task is episodic, starting with the first step of the Transformer and ending when the EOS token is sampled. The sequence of actions $A = a_1, a_2, \ldots, a_T$ during this episode represents the SMILES generated, and the product of the action probabilities $P(A) = \prod_{t=1}^{T} \pi(a_t \mid s_t)$ represents the model likelihood of the sequence formed.

We introduce a scoring function $S(A) \in [-1, 1]$ that rates the desirability of the sequences formed. The goal is to update the agent policy $\pi$ from the prior policy $\pi_{\text{Prior}}$ to increase the expected score for the generated sequences while remaining anchored to the prior policy. We define an augmented likelihood $\log P(A)_{\mathbb{U}}$ as a prior likelihood modulated by the desirability of a sequence:

$$\log P(A)_{\mathbb{U}} = \log P(A)_{\text{Prior}} + \sigma S(A)$$

where $\sigma$ is a scalar coefficient. The return $G(A)$ of a sequence $A$ can be seen as the agreement between the Agent likelihood $\log P(A)_{\mathbb{A}}$ and the augmented likelihood:

$$G(A) = -\left[\log P(A)_{\mathbb{U}} - \log P(A)_{\mathbb{A}}\right]^2$$

The goal of the Agent is to learn a policy which maximizes the expected return, achieved by minimizing the cost function $L(\Theta) = -G$. This approach can be described as a REINFORCE algorithm with a final step reward, allowing for effective optimization of the Agent's policy towards generating desirable molecular structures.

## E  ALGORITHMS

## F  EXPERIMENT SETTINGS

This metric, which we refer to as *AUC top-K*, is defined as:

Given a sequence of molecules $\{M_1, M_2, \ldots, M_N\}$ generated by a method, and an oracle function $O(M)$ that returns the property value of a molecule, the top-$K$ average property value at any point in the sequence is given by:

$$\text{Top-K Average}(M_1, M_2, \ldots, M_i) = \frac{1}{K} \sum_{j=1}^{K} O(M_{(j)}), \tag{12}$$

---

**Algorithm 1** REINVENT Transformer Pretraining Process

---

**Require:**

1: **function** PRETRAIN(restore_from=None)
2:     Initialize Vocabulary from file
3:     Load and preprocess data from 'ZINC' and 'ChEMBL'
4:     Filter and prepare the dataset
5:     Create a DataLoader for batch processing
6:     Initialize the Transformer model
7:     **if** restore_from is not None **then**
8:         Load saved model state
9:     **end if**
10:     Initialize optimizer with learning rate
11:     **for** each epoch **do**
12:         **for** each batch in DataLoader **do**
13:             Sample sequences (seqs) from DataLoader
14:             Compute log probability (log_p) with Transformer model
15:             Calculate loss: $loss = -\text{mean}(\text{log\_p})$
16:             Zero gradients
17:             Perform backpropagation
18:             Update model parameters
19:             **if** step%adjustment_interval == 0 **then**
20:                 Decrease learning rate by a specified factor
21:                 Sample a set of sequences for validation
22:                 Decode sampled sequences to SMILES
23:                 Validate the chemical structure of each SMILES
24:                 Calculate the percentage of valid SMILES
25:                 Display current epoch, step, loss, and % valid SMILES
26:             **end if**
27:         **end for**
28:         Save the current state of the Transformer model
29:     **end for**
30: **end function**
31: Call Pretrain function

---

where $M_{(j)}$ is the $j$-th highest property value molecule among the first $i$ molecules.

The *AUC top-K* is then the area under the curve when plotting the top-*K* average property value against the number of oracle calls up to molecule $M_i$, for $i = 1$ to $N$. This is calculated as:

$$\text{AUC top-K} = \int_1^N \text{Top-K Average}(M_1, M_2, \ldots, M_i)\, di \tag{13}$$

We set $K$ at 1, 10, and 100, capping the number of oracle calls at 10,000. All AUC values reported are min-max scaled to the range $[0, 1]$.

**Recall (Sensitivity):** Traditionally, recall is the proportion of actual positives correctly identified. In our context, it is the proportion of molecules with desirable properties (as judged by the oracle) that the method successfully identifies from the total 'N' molecules deemed desirable by the oracle.

**Precision (Positive Predictive Value):** Precision is the proportion of predicted positives that are true positives. Here, it is the proportion of molecules identified by the method as having desirable properties that are indeed validated by the oracle, out of the 'M' molecules selected by the method.

## G   EXPERIMENT RESULTS

## H   ABALATION STUDY

In Fig. 6, the AUC top10 curve for Mestranol Similarity is presented. Contrasted with the average accuracy curve, this AUC curve demonstrates a milder inclination initially, followed by a pronounced rise. Specifically, for the REINVENT-Transformer, the mean AUC top10 consistently surpasses that of REINVENT. Although the disparity is subtle during the initial oracle calls, it becomes more pronounced post the 5000th oracle call and remains so thereafter.

---

**Algorithm 2** REINVENT Transformer Optimization Process

---

**Require: Initialization**
1: Prior, Agent ← Transformer(Vocabulary)
2: Optimizer ← Adam(Agent.parameters, lr=config['learning_rate'])
3: Experience ← ExperienceReplay(Vocabulary)

    **Training Loop**
4: **while** True **do**
5:    **if** len(oracle) > 100 **then**
6:        Sort oracle buffer
7:        old_scores ← first 100 scores from oracle buffer
8:    **else**
9:        old_scores ← 0
10:    **end if**

    **Sampling and Evaluating Sequences**
11:    Seqs, AgentLikelihood, Entropy ← Agent.sample(config['batch_size'])
12:    UniqueIdxs ← Unique(Seqs)
13:    Seqs, AgentLikelihood, Entropy ← Seqs[UniqueIdxs], AgentLikelihood[UniqueIdxs], Entropy[UniqueIdxs]
14:    PriorLikelihood, - ← Prior.likelihood(Seqs)
15:    SMILES ← seq_to_smiles(Seqs, Vocabulary)
16:    Score ← Oracle(SMILES)
17:    **if** finish condition met **then**
18:        Break loop
19:    **end if**
20:    **if** len(oracle) > 1000 **then**
21:        Check for convergence based on new scores and old scores
22:        **if** convergence criteria met **then**
23:            Break loop
24:        **end if**
25:    **end if**

    **Loss Calculation**
26:    AugmentedLikelihood ← PriorLikelihood.float() + config['sigma'] × Score.float()
27:    Loss ← mean((AugmentedLikelihood - AgentLikelihood)$\hat{}$2)

    **Experience Replay (if enabled)**
28:    **if** config['experience_replay'] and len(Experience) > config['experience_replay'] **then**
29:        Experience replay steps
30:    **end if**

    **Optimization**
31:    Update experience with new experience
32:    LossRegularizer ← -mean(1 / AgentLikelihood)
33:    TotalLoss ← Loss + 5 × 10$\hat{}$3 × LossRegularizer
34:    Optimizer.zero_grad()
35:    TotalLoss.backward()
36:    Optimizer.step()
37:    Increment step counter
38: **end while**

---

The AUC top100 curve for Albuterol Similarity is illustrated in Fig. 7. In this context, the differential in performance between REINVENT-Transformer and REINVENT is more nuanced. It isn't until the 8000th oracle call that a discernible gap emerges. Ultimately, the REINVENT-Transformer exhibits marginally superior performance relative to REINVENT in this scenario.

| Method | Overview | Technical Details | Advantage | Disadvantage |
|---|---|---|---|---|
| REINVENT [32] | A method employing a policy-based reinforcement learning approach to instruct RNNs to produce SMILES strings. | Formulates molecular design as a Markov decision process with states representing partially generated molecules and actions as string manipulations. Rewards based on properties of interest. | Adaptable to generate other string representations like SELFIES. | Heavily reliant on the design of rewards. |
| Graph-GA [25] | A genetic algorithm that manipulates molecular representations using graphs, with graph matching and atom/fragment mutations. | Introduces crossover operations based on graph representations, unlike string-based genetic algorithms. | Offers a richer set of operations for exploring diverse chemical spaces. | Increased complexity due to graph-based operations. |
| SELFIES-REINVENT [14] | An extension of REINVENT for generating SELF-referencing Embedded Strings (SELFIES). | Uses a policy-based RL approach for SELFIES representation, ensuring syntactical validity. | Produces molecules with fewer syntactical errors. | Still dependent on reward system definition. |
| GP BO [38] | Combines Gaussian process Bayesian optimization with Graph-GA methods. | Leverages GP acquisition function integrated with Graph-GA techniques for sampling. | Balances exploration and exploitation effectively. | Higher computational costs due to GP and GA interplay. |
| STONED [31] | A modified genetic algorithm that manipulates tokens within SELFIES strings. | Interacts directly with tokens in SELFIES strings, differing from traditional string-based GAs. | Direct approach potentially reduces invalid chemical representations. | Limited to SELFIES, may not generalize to other representations. |
| SMILES-LSTM HC [8] | Iterative learning method using LSTM to understand the molecular distribution in SMILES strings. | Employs a variant of the cross-entropy method, fine-tuning the model with high-scoring molecules. | Iteratively refines the generative process. | Slow convergence if initial model is suboptimal. |
| SMILES-GA [8] | Genetic algorithm based on SMILES context-free grammar. | Implements genetic mutations and crossovers based on SMILES grammar. | Exploits SMILES structure for effective exploration. | Confined to SMILES grammar nuances, potentially missing novel structures. |
| SynNet [15] | Synthesis-based genetic algorithm operating on binary fingerprints and decoding to synthetic pathways. | Focuses on the synthesizability of generated molecules. | Prioritizes synthesizability, ensuring lab producibility of molecules. | Limited diversity in molecular space exploration due to synthesis emphasis. |
| DoG-Gen [7] | Tailored to learn the distribution of synthetic pathways. | Represents synthetic pathways as DAGs, using an RNN generator for modeling. Emphasizes synthesizability. | Structured approach to learning synthetic pathways. | Issues in capturing very long sequences with RNNs if not designed effectively. |
| DST [11] | Differentiable Scaffolding Tree method for molecular optimization using gradient ascent. | Abstracts molecular graphs into scaffolding trees, using a graph neural network for gradient estimation. | Direct optimization of molecular structures through gradient computation. | Possible loss of information due to abstraction to scaffolding trees. |

**Table 2: Summary of Methods in Molecular Design**

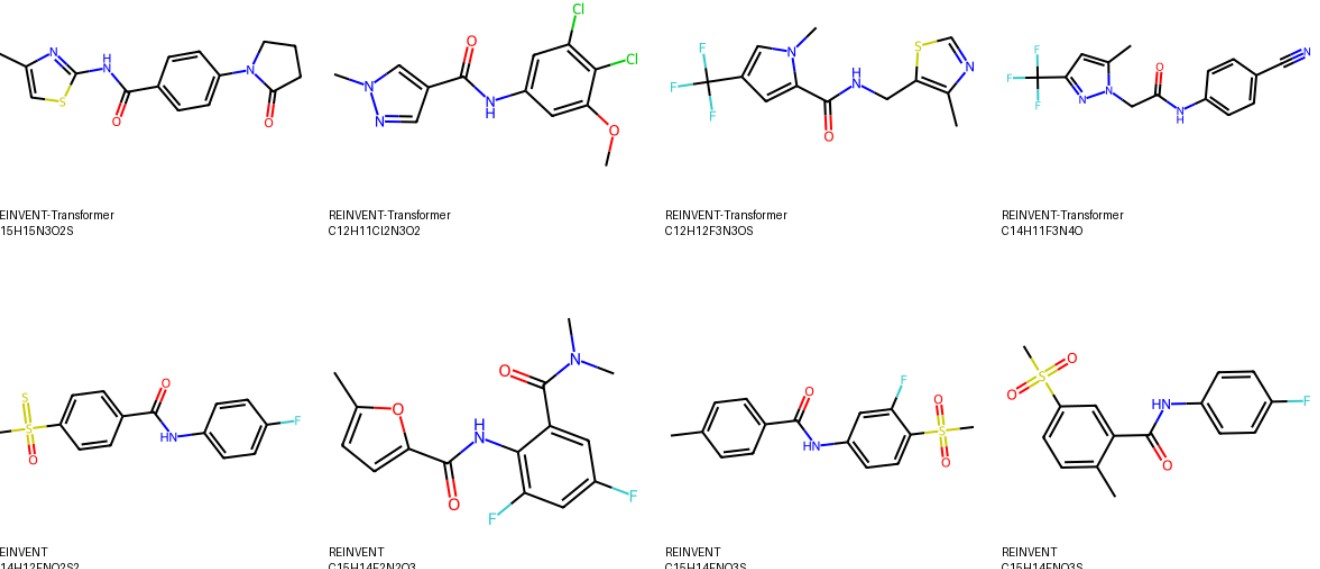

**Figure 5: Randomly selected SMILES chemical structures generated by the different models**

| Oracle | Model | Avg SA↓ | Diversity Top100 ↑ |
|---|---|---|---|
| Albuterol Similarity | REINVENT | 3.177 | 0.394 |
| | REINVENT-Trans | **3.173** | **0.408** |
| Amlodipine MPO | REINVENT | **3.478** | **0.391** |
| | REINVENT-Trans | 3.888 | 0.311 |
| Celecoxib Rediscovery | REINVENT | 3.458 | **0.551** |
| | REINVENT-Trans | **3.245** | 0.357 |
| DRD2 | REINVENT | **2.788** | **0.868** |
| | REINVENT-Trans | 2.914 | 0.464 |
| Deco Hop | REINVENT | 3.458 | **0.551** |
| | REINVENT-Trans | **3.240** | 0.457 |
| Fexofenadine MPO | REINVENT | 4.163 | 0.325 |
| | REINVENT-Trans | **4.113** | **0.411** |
| GSK3B | REINVENT | **3.146** | **0.884** |
| | REINVENT-Trans | **3.146** | **0.884** |
| Isomers C7H8N2O2 | REINVENT | 4.273 | 0.712 |
| | REINVENT-Trans | **2.589** | **0.796** |
| Isomers C9H10N2O2PF2Cl | REINVENT | 3.261 | 0.585 |
| | REINVENT-Trans | **3.245** | **0.686** |
| Median 1 | REINVENT | 4.571 | **0.408** |
| | REINVENT-Trans | **3.532** | 0.371 |
| Median 2 | REINVENT | **2.772** | **0.411** |
| | REINVENT-Trans | 2.877 | 0.389 |
| Mestranol Similarity | REINVENT | **3.799** | 0.267 |
| | REINVENT-Trans | 4.394 | **0.434** |
| Osimertinib MPO | REINVENT | **3.174** | **0.504** |
| | REINVENT-Trans | 3.799 | 0.447 |
| Perindopril MPO | REINVENT | 3.819 | **0.479** |
| | REINVENT-Trans | **3.766** | 0.357 |
| QED | REINVENT | **1.883** | **0.573** |
| | REINVENT-Trans | 3.422 | 0.540 |
| Ranolazine MPO | REINVENT | 3.468 | 0.421 |
| | REINVENT-Trans | **2.727** | **0.434** |
| Scaffold Hop | REINVENT | **2.857** | **0.555** |
| | REINVENT-Trans | 4.355 | 0.382 |
| Sitagliptin MPO | REINVENT | **2.639** | **0.692** |
| | REINVENT-Trans | 5.279 | 0.391 |
| Thiothixene Rediscovery | REINVENT | **2.899** | 0.373 |
| | REINVENT-Trans | 3.275 | **0.441** |
| Troglitazone Rediscovery | REINVENT | **3.275** | **0.441** |
| | REINVENT-Trans | 4.435 | 0.204 |
| Valsartan Smarts | REINVENT | 3.421 | 0.874 |
| | REINVENT-Trans | 3.421 | 0.874 |
| Zaleplon MPO | REINVENT | **1.991** | **0.614** |
| | REINVENT-Trans | 2.465 | 0.486 |

**Table 3: Avg SA and Diversity Top100**

| Model | SMILES | Score | Number |
|---|---|---|---|
| REINVENT-Transformer | Cc1csc(NC(=O)c2ccc(N3CCCC3=O)cc2)n1 | 0.9479 | 1656 |
| REINVENT-Transformer | COc1cc(NC(=O)c2cnn(C)c2)cc(Cl)c1Cl | 0.9477 | 1875 |
| REINVENT-Transformer | Cc1ncsc1CNC(=O)c1cc(C(F)(F)F)cn1C | 0.9475 | 1873 |
| REINVENT-Transformer | Cc1cc(C(F)(F)F)nn1CC(=O)Nc1ccc(C#N)cc1 | 0.9474 | 466 |
| REINVENT | CS(=O)(=S)c1ccc(C(=O)Nc2ccc(F)cc2)cc1 | 0.9481 | 6853 |
| REINVENT | Cc1ccc(C(=O)Nc2c(F)cc(F)cc2C(=O)N(C)C)o1 | 0.9481 | 5825 |
| REINVENT | Cc1ccc(C(=O)Nc2ccc(S(C)(=O)=O)c(F)c2)cc1 | 0.9481 | 4525 |
| REINVENT | Cc1ccc(S(C)(=O)=O)cc1C(=O)Nc1ccc(F)cc1 | 0.9481 | 4605 |

**Table 4: Randomly selected SMILES generated by the REINVENT and REINVENT-Transformer Models**

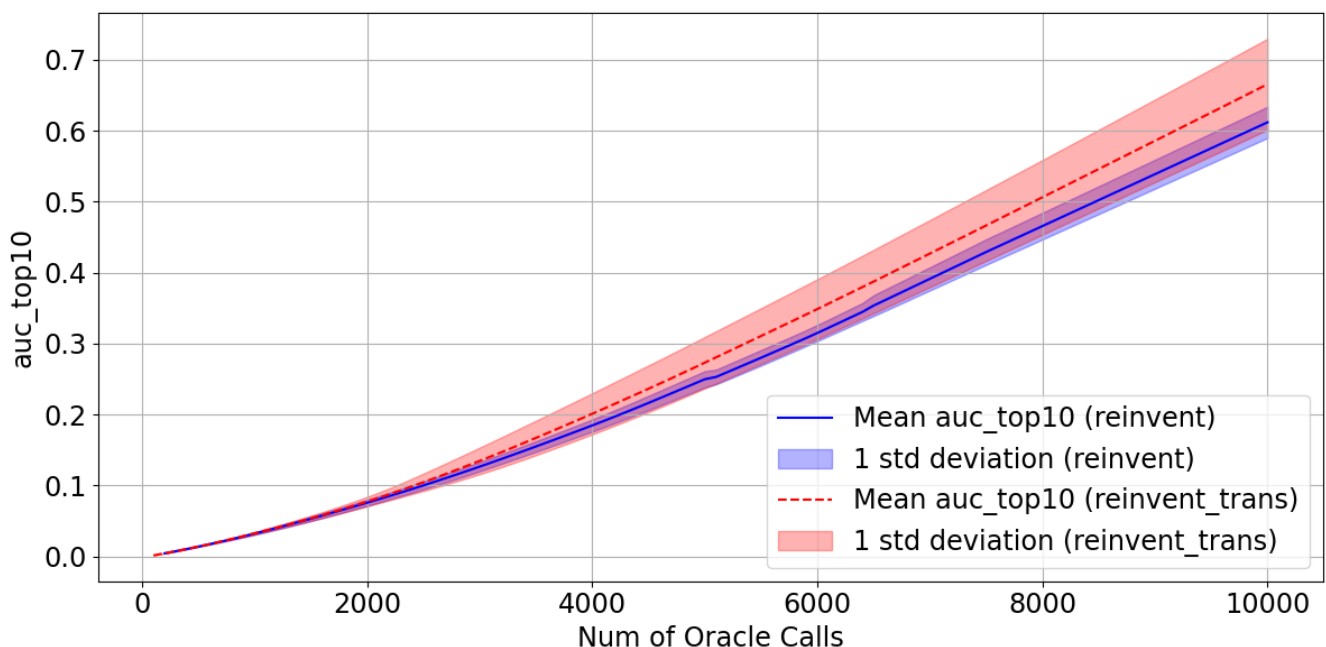

**Figure 6: Mean and Standard Deviation of auc_top10 over oracle calls for REINVENT and REINVENT-Transformer on oracle Mestranol_Similarity**

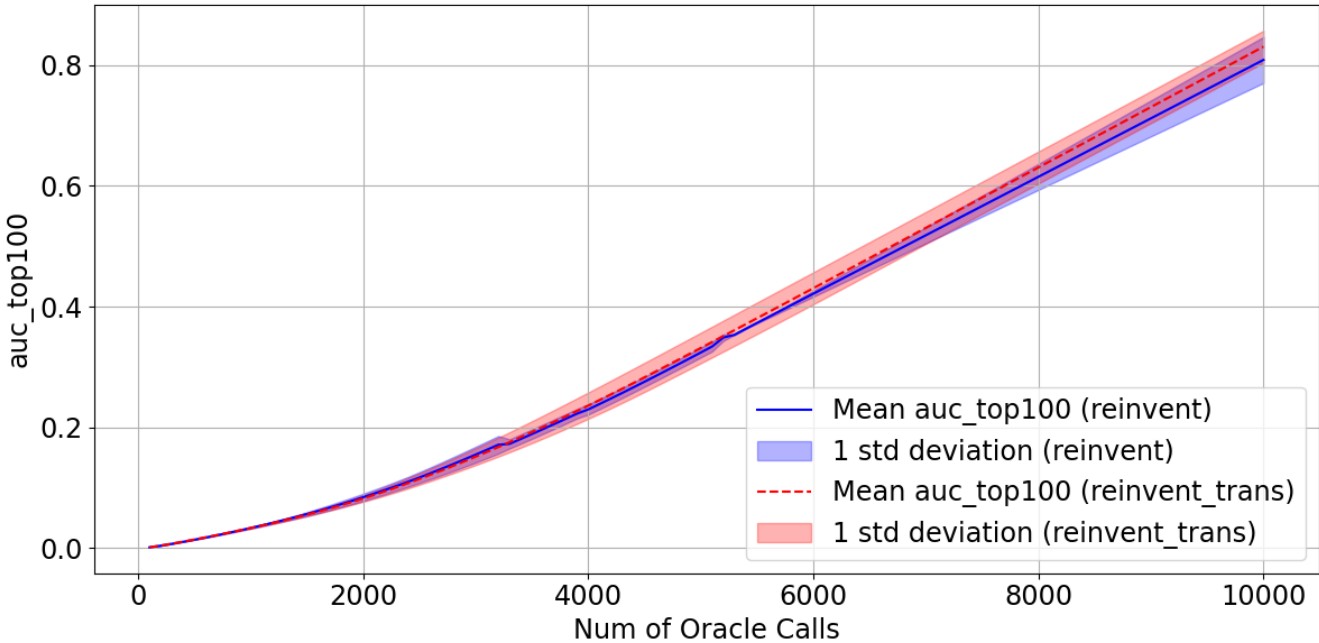

**Figure 7: Mean and Standard Deviation of auc_top100 over oracle calls for REINVENT and REINVENT-Transformer on oracle Albuterol_Similarity**