# OpenReview forum: "REINVENT-Transformer: Molecular De Novo Design through Transformer-based Reinforcement Learning"
_KDD.org/2024/Workshop/AIDSH — KDD-AIDSH 2024 Oral_

### Official Review · Reviewer_wmX8 · 2024-06-17
**Review of "REINVENT-Transformer: Molecular De Novo Design through Transformer-based Reinforcement Learning"**

**Rating:** 8
**Confidence:** 2

**Review:**

Summary
The manuscript integrates transformers with reinforcement learning, presenting a novel method for molecular design that enhances the efficiency and accuracy of molecular structure generation. Additionally, the manuscript demonstrates the model's potential and superior performance across a variety of molecular design tasks through a series of experiments. The work is solid and well-executed.

Strengths
- The manuscript successfully applies a framework combining transformers with reinforcement learning fine-tuning to the field of molecular design, showcasing an exemplary AI for science application.
- The inclusion of case studies provides a clear demonstration of the model's interpretability, which significantly aids in understanding the practical implications of the research.

Weaknesses
- The manuscript could improve its formatting. For example, in the "Related Works" section, the text about transformers exceeds the text box limits, and in Section 3.2, the citation markers extend beyond the line width.

Question
- I am not very familiar with the field of chemical molecules, but I wonder why transformers might even surpass graph-based methods in tasks involving molecules, which naturally form graph structures. Could the authors provide some insight into this?

---

### Official Review · Reviewer_T58k · 2024-06-18
**Advanced Approach**

**Rating:** 8
**Confidence:** 3

**Review:**

The paper presents a cutting-edge application of Transformer-based models for molecular de novo design. The ability of Transformers to handle long-term dependencies in molecular sequences marks a substantial improvement over traditional RNN-based models.The research provides an in-depth comparison with existing methods, showcasing the superior performance of the REINVENT-Transformer in generating molecules with desired properties. This comprehensive benchmarking against various oracles highlights the model’s efficacy. The practical applications of the REINVENT-Transformer in drug discovery are well articulated. Its ability to generate compounds with high predicted activity against biological targets and perform tasks like scaffold hopping and library expansion is highly valuable.The experimental setup and evaluation are robust, with clear improvements demonstrated over baseline methods. The use of multiple metrics, such as AUC for top-K property values, provides a well-rounded assessment of the model’s performance. The REINVENT-Transformer sets a new standard in molecular de novo design. Its success in leveraging Transformers for this task suggests that further developments and refinements could lead to even greater advancements in drug discovery.
Limitations: The model’s predictions are based on computational evaluations. Experimental validation of the generated compounds’ properties in a laboratory setting is crucial to confirm their real-world applicability. The model’s ability to handle highly diverse molecular structures, particularly those with unconventional features, is not fully explored. Future work could focus on enhancing its generalizability across different types of molecules.

---

### Decision · Program_Chairs · 2024-06-28

Accept (Oral)